# POST-TRAINING QUANTIZATION OF VISION ENCODERS NEEDS PREFIXING REGISTERS

## ABSTRACT

Transformer-based vision encoders—such as CLIP—are central to multimodal intelligence, powering applications from autonomous web agents to robotic control. Since these applications often demand real-time processing of massive visual data, reducing the inference cost of vision encoders is critical. Post-training quantization offers a practical path, but remains challenging even at 8-bit precision due to massive-scale activations (i.e., outliers). In this work, we propose *RegCache*, a training-free algorithm to mitigate outliers in vision encoders, enabling quantization with significantly smaller accuracy drops. The proposed RegCache introduces outlier-prone yet semantically meaningless prefix tokens to the target vision encoder, which prevents other tokens from having outliers. Notably, we observe that outliers in vision encoders behave differently from those in language models, motivating two technical innovations: middle-layer prefixing and token deletion. Experiments show that our method consistently improves the accuracy of quantized models across both text-supervised and self-supervised vision encoders.

## 1 INTRODUCTION

Transformer-based vision encoders, such as CLIP or DINOv2, lie at the core of modern multimodal intelligences (Radford et al., 2021; Oquab et al., 2024). Leveraging the scalability of vision transformer (ViT) backbone (Dosovitskiy et al., 2021), these models can be pretrained with massive amount of data and computation, yielding highly informative and versatile visual features. Today, vision encoders are now being adopted as plug-and-play components across diverse multimodal applications, ranging from autonomous web agents to robotic control (Palanisamy et al., 2025).

Lowering the inference cost of vision encoders is essential, as their applications often require real-time processing of visual signals on edge devices, e.g., on-device robotic control (Kim et al., 2025). Post-training quantization (PTQ) is a promising solution for this purpose, as the technique can substantially reduce the memory and computation burden of the models without any additional training (Choukroun et al., 2019). In particular, the activation quantization of vision encoders is of significant importance; the models are typically non-autoregressive and run on edge hardwares, and thus more likely to be in compute-bound scenarios than in memory-bound one.[1] Quantizing both activations and weights enables replacing high-precision matrix multiplications with low-precision operations, e.g., int8, effectively reducing the computation and energy needed.

However, quantizing the activations of vision encoders is challenging due to the *outlier* activations, i.e., few activations with extremely large magnitude. In particular, large-scale vision encoders tend to have outliers emerging in a small number of channels at the middle-to-final blocks of the models (Sun et al., 2024). Outliers force the quantization range of the activations to be much larger than usual, leading to a significant quantization error. Strategies for outlier-robust quantization have been actively studied, particularly in the context of large language models (LLMs), which also suffers from outliers similarly (Dettmers et al., 2022; Xiao et al., 2023; Lin et al., 2024). These approaches, however, typically involve applying different precision or quantization range to different tokens or channels. Such operations require much implementation and computational overhead, and are difficult to be applied for static activation quantization (Son et al., 2024; Chen et al., 2024).

---

[1]This is in contrast with the scenarios of autoregressive large language models running on GPU servers. The inference is memory-bound and the weight-only quantization becomes effective.

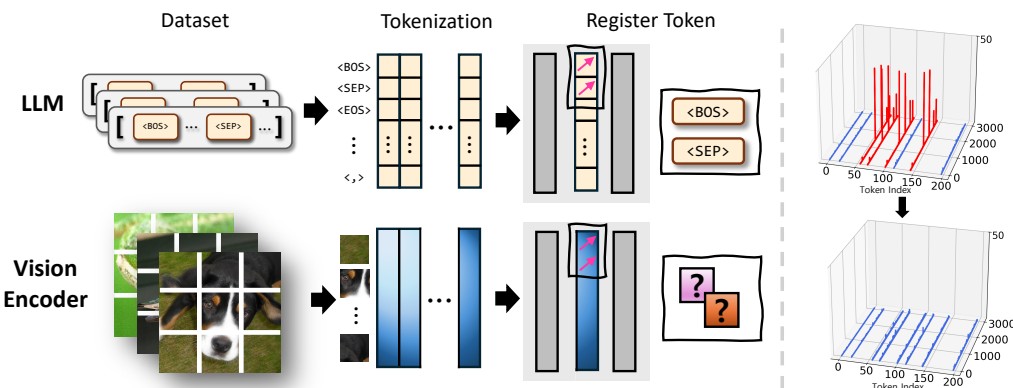

Figure 1: **(Left) Sink tokens in LLMs vs. vision encoders.** In LLMs, well-known sink tokens exist in a closed-set vocabulary. In contrast, vision encoders take image inputs composed of diverse patches that are continuously mapped into an embedding space, making the discovery of sink tokens more challenging. **(Right) Activation magnitudes in CLIP-B/16, with and without RegCache.** RegCache discovers and inserts sink to quantization-sensitive layers, not as an input. This operation mitigates outliers, thereby narrowing the dynamic range and enabling more effective activation quantization under low bitwidths.

An emerging alternative approach is to directly mitigate outliers in the model by prefixing *attention sink* tokens, i.e., semantically meaningless tokens—such as ⟨BOS⟩ or ⟨SEP⟩—which collect large attention from other tokens (Xiao et al., 2024; Sun et al., 2024). Recent studies on LLM quantization observe that adding the activations of these sink tokens as a prefix in each attention layer can dramatically reduce the activation magnitudes of other tokens, thereby boosting the post-quantization accuracy of large language models (Yang et al., 2024a; Son et al., 2024; Chen et al., 2024).

Naturally, one may ask: Can we mitigate the outliers in vision encoders by prefixing attention sinks? Unfortunately, it remains unclear which vision encoder token can play an analogous role to attention sinks in language models. Unlike language models, most existing vision encoders are not pretrained with tokens that are designated to be semantically meaningless. A recent line of works report that adding such meaningless tokens—called "registers"—during the training provide meaningful advantages in term of the interpretability of vision transformers (Darcet et al., 2024). However, it is still a rare practice to include such registers for vision encoders (Fig. 1, left).[2]

**Contribution.** In this work, we introduce *RegCache* (Register Caching), a novel prefix-based outlier mitigation algorithm for quantizing pretrained vision encoders. The method is inspired by the following empirically observed phenomenon on the emergence of registers in vision encoders:

"While no sink token exists at the first layer of vision encoders, sink tokens emerge gradually in the middle layers, giving rise to the outliers. Moreover, such tokens are highly similar across images, and thus can work as a universal middle-layer register for any input image at test phase."

Based on this observation, RegCache mitigates outliers by discovering and prefixing these middle-layer registers to the target vision encoder, in the form of a pre-computed key-value cache. Importantly, unlike in LLM prefixing (Son et al., 2024), the tokens are prefixed only for middle-to-final layers, and do not affect the early layers. Furthermore, RegCache additionally deletes out the tokens that have gradually became attention sink tokens, which are likely to suffer from outliers in subsequent blocks. By such adding-and-deleting of tokens, RegCache replaces internally emerging sink tokens with external pre-computed caches, so that sink tokens do not affect the activation quantization range of the model. The whole procedure does not require any further training of the vision encoder, rendering RegCache a versatile and easy-to-use method.

Throughout our experiments, we apply the proposed RegCache to a wide range of text-supervised and self-supervised vision encoders, combining it with several recent PTQ techniques for vision transformers. We observe that RegCache improves the prediction accuracy of the quantized vision encoder quite consistently over all setups considered.

---

[2]In this regard, DINOv3 is a pleasant exception (Siméoni et al., 2025).

## 2 RELATED WORK

**Outlier in large-scale transformers.** In large-scale transformers, it has been observed that some activation magnitudes at certain layers become significantly larger than others; this phenomenon is referred to as the emergence of outliers (Kovaleva et al., 2021; Timkey & Van Schijndel, 2021; Bondarenko et al., 2021; Dettmers et al., 2022). Sun et al. (2024) conducts a systematic study of outliers, to show that they arise due to the softmax in the self-attention mechanism in LLMs and ViTs. They show that certain tokens in LLMs, e.g., ⟨BOS⟩ or ⟨SEP⟩, consistently exhibit extreme activation magnitudes. Several works in the vision domain show that outlier tokens in ViTs typically correspond to uninformative uniform background patches, and that removing them can improve internal representations (Darcet et al., 2024; Jiang et al., 2025; Lu et al., 2025). In contrast to LLMs, it remains unclear which "specific" visual tokens give rise to outliers in ViT-based vision encoders, given different characteristics between language and image data (e.g., different images have different backgrounds). In this work, we find that, across a wide range of vision encoders, outlier tokens typically emerge in intermediate blocks and exhibit similar features across images, enabling them to be pre-computed for a given use-case, such as post-training quantization.

**Improving vision transformers via attention sink control.** Attention sinks, first highlighted by Xiao et al. (2024), are tokens with little or no semantic information that nevertheless attract excessive attention in both LLMs (Guo et al., 2024) and ViTs (Darcet et al., 2024). In ViTs, critically, these sink tokens act as noise in the attention map, hindering the model's ability to capture relations between different patches and thereby degrading downstream visual performance (Darcet et al., 2024; Jiang et al., 2025; Kang et al., 2025; Lu et al., 2025). Seminal work by Darcet et al. (2024) adds register tokens that absorb much attention and thus mitigate attention sinks during training. More recently, Jiang et al. (2025) suggest identifying register neurons (i.e., specific channels in the linear layers of ViT blocks) before training and "delete-and-paste" the maximum value of register neurons to zero-initialized token at test time. Taking a slightly different perspective, we ask how to leverage sink tokens to improve PTQ. From this PTQ-oriented perspective, our core strategy is to remove sink tokens that are prefixed (and also pre-computed) at test time.

**Post-training quantization methods for vision transformers.** There have been a lot of effort to reduce the inference cost of large-scale ViT-based models via PTQ (Yang et al., 2024b; Wu et al., 2025a;b). Early PTQ methods for ViTs address quantization errors by assigning dynamic bitwidths to self-attention-sensitive layers (Liu et al., 2021). Subsequent studies identify that low PTQ performance of ViTs are due to outliers, originating from operations such as LayerNorm, softmax, and GELU activation. RepQ-ViT (Li et al., 2023) and PTQ4ViT (Yuan et al., 2022) propose novel quantization schemes to isolate and minimize the impact of outliers. Methods such as NoisyQuant (Liu et al., 2023) attempt to alleviate the tail-like behavior of activations by adding noise or reshaping distributions. By contrast, our approach handles outliers via token prefixing rather than directly controlling quantizer granularity, and it can be easily integrated with existing methods easily.

## 3 A CLOSER LOOK AT OUTLIERS IN VISION ENCODERS

Outliers in vision encoders emerge at seemingly random background patch tokens in given images, while in LLMs they tend to appear at specific location or types of tokens (Sun et al., 2024; Darcet et al., 2024). Such lack of specificity makes it difficult to mitigate the outliers in vision encoders through prefixing (Son et al., 2024), as the strategy requires caching the tokens that induce outliers universally across any input that can be given to the model at the test time (i.e., registers).

In this section, we provide two observations suggesting that an alternative strategy may be effective: "Find universal sink tokens in middle layers of the model, and prefix them in these middle layers."

- Section 3.1: Layerwise quantization sensitivity of vision encoders is particularly high in the middle layers where outliers emerge, and low otherwise—thus, prefixing need not be done in early layers.
- Section 3.2: In the middle layers where outliers begin emerging, the cosine similarity of the outlier tokens across different images become very high—thus, they can work as universal registers.

In Section 3.3, we report an intriguing observation which sheds light on why such late emergence of outliers happen in vision encoders. Precisely, the observation suggest that: "vision encoders require early layers to process the image, in order to understand which tokens are semantically meaningless."

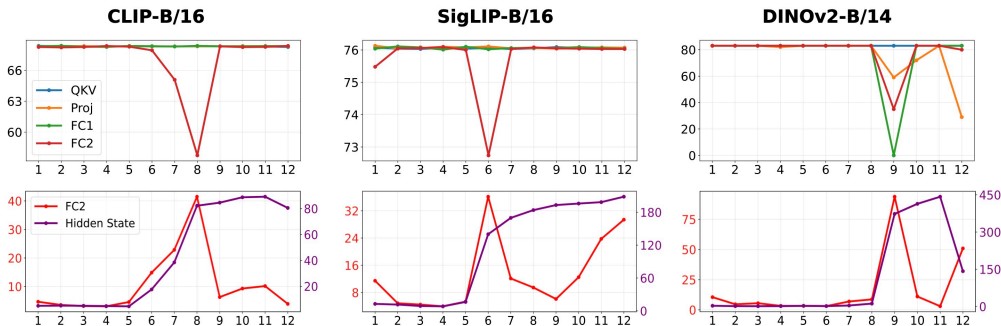

Figure 2: **(Top) Layerwise quantization sensitivity (%).** Zero-shot ImageNet-1k accuracy when we quantize only one layer to W8A8. **(Bottom) Layerwise max token norms.** The largest $\ell_\infty$-norm of all tokens in an image, averaged over the ImageNet-1k validation set.

### 3.1 LAYERWISE QUANTIZATION SENSITIVITY AND OUTLIERS

We first analyze the quantization sensitivity of each layers in vision encoders, and establish connection to the emergence of outliers in hidden states (i.e., block outputs).

In Fig. 2 (top), we report the zero-shot ImageNet-1k accuracy of various vision encoders, when we quantize different layers of a single transformer block to W8A8. We observe that quantization-sensitive layers—i.e., layers with substantial accuracy drop when quantized—are highly localized to the MLP projection layers in one or two middle layers. In DINOv2, the performance degradation is the sharpest, and takes place in other layers and blocks as well. Furthermore, as can be seen from Fig. 2 (bottom), these quantization-sensitive layers coincide with the layers where the activation outliers in the hidden states (i.e., block output + residual) begin to emerge. Together, the plots suggest that outliers are indeed the driving factor of performance drops when quantizing vision encoders.

These observations extend and refine the observations of prior works that vision encoders have high-norm tokens in middle-to-later layers (Darcet et al., 2024; Jiang et al., 2025). In particular, we establish a concrete connection between the quantized accuracy and the high-norm behaviors. Furthermore, our findings suggest that FC2 activations or measuring quantization sensitivity directly may be useful in pinpoint the block where prefixing should be conducted. We provide the results on other vision encoders (OpenCLIP and SigLIP2) in Appendix B.

### 3.2 UNIVERSALITY OF OUTLIER TOKENS

Next, we take a closer look at the outlier tokens in the quantization-sensitive layer. In particular, we measure the cosine similarity of the middle-layer outlier tokens (i.e., having the largest $\ell_\infty$-norm) collected for two images. We have used 64 randomly sampled images from the validation split of ImageNet-1k, and computed the average pairwise similarity.

Table 1: Average cosine similarity between tokens in SigLIP-B/16.

| Token Type | Cosine sim. |
|---|---|
| Normal tokens | 0.26 (±0.10) |
| Outlier tokens | 0.89 (±0.07) |

From Table 1, we observe that the outlier tokens are highly similar across different images, having the mean cosine similarity of 0.89. On the other hand, the non-outlier tokens are much more dissimilar from each other, with 0.26 mean cosine similarity. This indicates that outliers share components that are largely independent of input image, and thus may represent universal features that persist across samples.

### 3.3 WHY THE MIDDLE LAYERS?

Why do outliers in vision encoders emerge in middle layers, while the outliers in LLM emerges in the early layers? We hypothesize that this is essentially due to the fact that it is not readily clear from the raw image tokens which are *semantically meaningless*—they become clearer after being processed by first several blocks of the vision encoder. This is in contrast with the case of LLMs, where some tokens are clearly meaningless even at the first glance, e.g., $\langle \text{BOS} \rangle$, $\langle \text{SEP} \rangle$.

To validate this hypothesis, we design an experiment where we can compare the emergence of outliers for images where some patches are clearly meaningless, against those where the distinction is less clear. Precisely, using the test set of ImageNet-9 (Xiao et al., 2021), we compare the outliers of the foreground-only images—where the background pixels are zero-ed out—to the vanilla images. In principle, in foreground-only images, the semantically meaningless patches should be identifiably more easily, i.e., with a smaller number of blocks for processing.

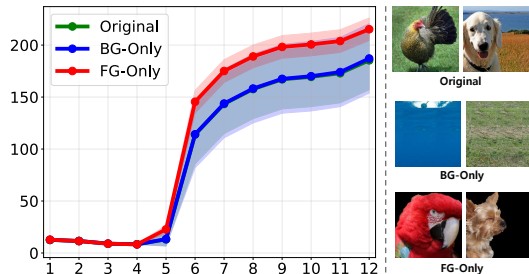

Figure 3: Emergence of outliers in foreground- and background-only images on SigLIP-B/16.

In Figure 3, we observe that the outliers in foreground-only images indeed emerge earlier and larger than in original, supporting our hypothesis. In contrast, if we remove the foreground and keep the background only, the outlier behaviors did not change noticeably. In Appendix G, we provide additional analysis on the case for the vision encoders trained with registers. There, we observe that models trained with registers behave similarly to LLMs, with outliers emerging from early layers.

## 4 METHOD

Before describing our outlier mitigation method, recall two observations from the Section 3:

- Outliers in vision encoders tend to emerge in the middle layers of the model (e.g., fully-connected layers in block 5–8), whereas for LLMs outliers emerge from the early layers.
- Sink tokens (i.e., outlier-prone) discovered in the middle layers tend to be similar across images.

Putting these together with prior observations in LLMs—prefixing additional sink tokens mitigate outliers in other tokens (Son et al., 2024)—we arrive at the following hypothesis:

> *"Middle-layer sink tokens from another image can play a role similar to registers, and thus can help mitigating outliers in vision encoders."*

Based on this hypothesis, we propose RegCache (Register Caching), an outlier mitigation algorithm which replaces internally emerging sink tokens by inserting sink tokens discovered from reference images as registers. In a nutshell, RegCache operates in three steps (see Fig. 4).

(1) **Curating** a set of register candidate tokens from a pool of reference images.     ▷ Section 4.1

(2) **Caching** the keys and values of selected register candidates to the sensitive layers. ▷ Section 4.2

(3) **Deleting** internally emerging sink tokens to clean up remaining outliers.     ▷ Section 4.3

The proposed RegCache does not involve any training or tuning of the model, and only conducts several rounds of validation on some reference task and data (will be discussed below). Thus, the algorithm does not require an excessive amount of training data or computational resource. At the inference phase, the algorithm adds and removes several tokens, which slightly affects the computational cost of the model. However, as we will see in Appendix E, the change is negligible.

### 4.1 CURATING

Given a pretrained vision encoder, we first identify the quantization-sensitive layer of the model. Then, we construct a curated set of candidate register tokens by selecting top-$k$ tokens with largest sensitive-block activations, among all tokens taken from a pool of reference images.

**Identifying the quantization-sensitive layer.** As in Section 3.1, we quantize each layer of the given vision encoder separately, and select the one that leads to the lowest accuracy on some reference task as the quantization-sensitive layer. Here, if we have a specific base quantization algorithm in mind (to be used after the outlier mitigation), we can use the algorithm; otherwise, we will use the vanilla round-to-nearest quantization. The reference task is selected as the task that is considered representative of visual understanding, *e.g.*, classification on the ImageNet-1k training set.

Figure 4: **Overview of the proposed method.** We first identify a universal register by analyzing the inputs of quantization-sensitive layers across blocks. During inference, the register is inserted into each block, and outlier tokens are removed from the most quantization-sensitive blocks.

**Curating the set of register candidates.** After identifying the quantization-sensitive layer, we construct the set of register candidate tokens, which are likely to play the role of a register when inserted to the quantization-sensitive layer. This is done by predicting on the a pool of reference images with the vision encoders and selecting tokens with the largest $\ell_\infty$ norm at the quantization-sensitive layer.

More formally, let $l_q$ denote the quantization-sensitive layer of the target vision encoder, and let $\Phi_l(\mathbf{x})$ denote the set of tokens at the input of the $l$-th layer of the vision encoder, given an image $\mathbf{x}$. Then, we can construct the set of register candidates as follows.

$$\mathcal{S} = \operatorname{argtopk}\left\{\|\mathbf{z}\|_\infty \;\middle|\; \mathbf{z} \in \Phi_{l_q}(\mathbf{x}), \quad \text{for some } \mathbf{x} \in \mathcal{I}_{\text{ref}}\right\}, \tag{1}$$

where $\mathcal{I}_{\text{ref}}$ denotes the pool of reference images. In this paper, we use the 50,000 randomly drawn images from the training split of the ImageNet-1k dataset as this pool of images, and let $k = 20$.

As the sink tokens often emerge a few blocks before the quantization-sensitive layer, we also conduct a similar search for several blocks preceding the quantization-sensitive layer. We search up to three additional preceding blocks to construct register candidate sets for each block.

## 4.2 CACHING

Having the set of register candidates $\mathcal{S}$ constructed, we now determine the register $\mathbf{z}^* \in \mathcal{S}$ and the number of repetitions $\tau^* \in \mathbb{N}$ of the register to be inserted to the target vision encoder, in the form of a key-value cache. The determination of $(\mathbf{z}^*, \tau^*)$ is done by grid search, using the accuracy of the quantized vision on some reference task. Precisely, the search is done as follows.

- First, we compute key-value caches for each register candidate token $\mathbf{z} \in \mathcal{S}$, for the quantization-sensitive layer and all succeeding blocks using the *unquantized* vision encoder.
- Then, we insert each KV cache to the *quantized* vision encoder separately, with various number of repetitions. We vary the number of repetitions $\tau$ within the range $\{1, 2, \ldots, 15\}$.
- Lastly, we evaluate the performance of each combination of $(\mathbf{z}, \tau)$ on the reference task, and select the pair with the maximum reference task accuracy. In other words, we select:

$$(\mathbf{z}^*, \tau^*) = \arg\max\left\{\operatorname{acc}_{\text{ref}}(\mathbf{z}, \tau) \mid \mathbf{z} \in \mathcal{S}, \tau \in \{1, \ldots, 15\}\right\} \tag{2}$$

where $\operatorname{acc}_{\text{ref}}$ denotes the accuracy on the reference task. Here, as in Section 4.1, we consider classification on the training split of ImageNet-1k dataset as our reference task.

## 4.3 DELETING

Finally, we add a *token deletion layer* at the input of the quantization-sensitive block (*i.e.*, where $l_q$ is located) of the vision encoder. At the inference phase, this layer removes the sink tokens that emerge among the image patch tokens, thus removing any remaining outliers. Precisely, given some test image $\mathbf{x}_{\text{test}}$, the layer selects the tokens with the top-$\tilde{k}$ $\ell_\infty$ norm, *i.e.*,

$$\mathcal{D} = \operatorname{argtop}\tilde{k}\left\{\|\mathbf{z}\|_\infty \;\middle|\; \mathbf{z} \in \Phi_{l_q}(\mathbf{x}_{\text{test}})\right\} \tag{3}$$

and remove these tokens from the model. Here, similarly to the curating and caching steps, the number of tokens to be removed—*i.e.*, $\tilde{k}$—is tuned using the reference task.

Table 2: **Zero-shot classification accuracy of various vision encoders on ImageNet-1k.** We have used various base quantization algorithms to quantize to 8/6bits. The best results are marked in **bold**. We do not plot the case of 6-bit naïve quantization, which results in near-zero accuracy. Best/Average $\Delta$ denote the gap between the best/average performances with and without RegCache.

| Method | CLIP-B/16 | | OpenCLIP-B/16 | | SigLIP-B/16 | | SigLIP2-B/16 | | DINOv2-B/14 | |
|---|---|---|---|---|---|---|---|---|---|---|
| | W8A8 | W6A6 | W8A8 | W6A6 | W8A8 | W6A6 | W8A8 | W6A6 | W8A8 | W6A6 |
| FP32 | 68.32 | | 70.22 | | 76.05 | | 78.47 | | 83.26 | |
| Naïve | 34.01 | – | 46.12 | – | 69.71 | – | 26.04 | – | 19.20 | – |
| w/ RegCache | 61.44 | – | 67.14 | – | 74.42 | – | 68.65 | – | 22.07 | – |
| PTQ4ViT | 67.69 | 51.60 | 69.39 | 59.98 | 75.57 | 68.68 | 76.92 | 41.54 | **82.97** | 78.28 |
| w/ RegCache | **67.78** | 58.19 | **69.70** | 64.09 | 75.82 | 70.97 | **77.26** | 66.15 | 82.93 | **79.09** |
| RepQ-ViT | 67.39 | 53.25 | 68.70 | 46.51 | 75.23 | 73.32 | 76.43 | 64.91 | 82.27 | 19.53 |
| w/ RegCache | 67.34 | **66.65** | 68.51 | 46.01 | 75.21 | **73.69** | 76.51 | **70.88** | 81.72 | 23.02 |
| NoisyQuant | 63.20 | 46.19 | 67.08 | 59.05 | 75.50 | 71.10 | 70.83 | 44.50 | 71.46 | 49.25 |
| w/ RegCache | 67.62 | 59.48 | 69.60 | **67.48** | **75.88** | 73.03 | 76.35 | 62.58 | 69.40 | 47.22 |
| Best $\Delta$ | +0.09 | +13.40 | +0.21 | +7.50 | +0.31 | +0.37 | +0.34 | +5.97 | -0.04 | +0.81 |
| Average $\Delta$ | +7.97 | +11.09 | +5.92 | +4.01 | +1.33 | +1.53 | +12.14 | +16.22 | +0.06 | +0.76 |

## 5 EXPERIMENTS

In this section, we first describe the experimental setup in Section 5.1. Next, in Section 5.2, we show that for zero-shot image classification and image–text retrieval tasks, combining RegCache with either naïve quantization or other baseline algorithms yields a significant performance gain. Furthermore, we analyze the behavior of token norm outliers, assess the generalization ability of the pre-computed prefix tokens, and validate our design choices through ablation studies.

### 5.1 SETUP

**Vision encoders.** We evaluate the proposed method on total five widely used vision encoders: (1) CLIP (Radford et al., 2021), (2) OpenCLIP (Cherti et al., 2023), (3) SigLIP (Zhai et al., 2023), (4) SigLIP2 (Tschannen et al., 2025), (5) DINOv2 (Oquab et al., 2024).

These models were selected to cover diverse setups. Specifically, CLIP and SigLIP are trained on image–text pairs with contrastive objectives, whereas DINOv2 is trained on image-only datasets. Regarding input tokens, CLIP and DINOv2 use a class token to extract global features, while SigLIP and SigLIP2 use patch-wise pooling to generate a token that captures global information.

**Evaluation.** We evaluate the quality of the quantized vision encoders by measuring the zero-shot accuracy on two downstream tasks: (1) image classification on ImageNet-1k (Deng et al., 2009) and (2) text-image retrieval on MS-COCO (Lin et al., 2014). To further validate the generalizability of the prefix searched from zero-shot classification, evaluation is conducted on a diverse set of image classification benchmarks, including Stanford Cars (Krause et al., 2013), Flowers-102 (Nilsback & Zisserman, 2008), Food-101 (Bossard et al., 2014), and CIFAR-100.

**Base quantization algorithms and details.** To assess broad applicability, we evaluate two baseline strategies for activation quantization in vision transformers: (1) module-specific quantizer designs that mitigate activation-distribution bottlenecks and and (2) input-side distribution shaping to reduce quantization error. Specifically, we use PTQ4ViT (Yuan et al., 2022) and RepQ-ViT (Li et al., 2023) for (1), and NoisyQuant (Liu et al., 2023) for (2). We provide further discussions in Appendix A.

Each baseline adopts per-tensor dynamic quantization with 8-bit and 6-bit precision, using 1,024 and 32 calibration samples and for NoisyQuant and RepQ-ViT, respectively, as in original papers. Additionally, for CLIPs and SigLIPs, prefixes are inserted from the searched layer to the final layer. As a self-supervised model, DINOv2 exhibits distinct characteristics compared to text-supervised counterparts. Accordingly, inserting the prefix only into the searched layer yields better results. A deeper investigation into this behavior is left as future work.

Table 3: Zero-shot image–text retrieval performance of CLIP and SigLIP on MS-COCO. The best results, in both recall at 1 and recall at 5, are marked in **bold**. Best/Average $\Delta$ denote the gap between the best/average performances with and without RegCache.

(a) CLIP-B/16

| | I $\rightarrow$ T | | T $\rightarrow$ I | |
| --- | --- | --- | --- | --- |
| | R@1 | R@5 | R@1 | R@5 |
| FP32 | 52.94 | 77.78 | 32.73 | 57.70 |
| Naïve | 22.76 | 41.92 | 14.08 | 30.43 |
| w/ RegCache | 47.78 | 73.56 | 29.47 | 54.04 |
| PTQ4ViT | 52.78 | **77.94** | 32.00 | 56.65 |
| w/ RegCache | **53.22** | 77.60 | **32.42** | **57.18** |
| RepQ-ViT | 44.52 | 68.64 | 23.01 | 45.20 |
| w/ RegCache | 44.94 | 68.24 | 22.93 | 45.30 |
| NoisyQuant | 48.94 | 74.10 | 31.07 | 56.18 |
| w/ RegCache | 49.84 | 75.06 | 30.36 | 54.99 |
| Best $\Delta$ | +0.44 | -0.34 | +0.42 | +0.53 |
| Average $\Delta$ | +6.70 | +7.97 | +3.76 | +5.76 |

(b) SigLIP-B/16

| | I $\rightarrow$ T | | T $\rightarrow$ I | |
| --- | --- | --- | --- | --- |
| | R@1 | R@5 | R@1 | R@5 |
| FP32 | 67.68 | 86.94 | 47.19 | 72.46 |
| Naïve | 60.04 | 82.66 | 41.80 | 67.40 |
| w/ RegCache | 65.38 | 85.78 | 46.25 | 71.20 |
| PTQ4ViT | 66.86 | 87.02 | 47.16 | 71.94 |
| w/ RegCache | **67.30** | 86.86 | **47.26** | **72.38** |
| RepQ-ViT | 65.90 | 86.78 | 46.33 | 71.48 |
| w/ RegCache | 65.90 | 86.74 | 46.71 | 71.66 |
| NoisyQuant | 67.10 | 86.96 | 46.76 | 72.05 |
| w/ RegCache | 67.02 | **87.16** | 46.98 | 72.09 |
| Best $\Delta$ | +0.20 | +0.14 | +0.20 | +0.33 |
| Average $\Delta$ | +1.43 | +0.78 | +1.29 | +1.12 |

Table 4: **Reduction in maximum token norm within quantization-sensitive layers in W8A8.** Both the maximum token norm and the average norm of other tokens are reported. We report mean across 1,000 image samples.

| Model | Max token | | Other tokens | |
| --- | --- | --- | --- | --- |
| | **Vanilla** | **w/ RegCache** | **Vanilla** | **w/ RegCache** |
| CLIP | 61.17 | 15.30 | 10.47 | 8.67 |
| OpenCLIP | 122.99 | 12.38 | 11.22 | 9.04 |
| SigLIP | 78.09 | 12.15 | 9.85 | 9.54 |
| SigLIP2 | 244.78 | 30.45 | 8.97 | 8.86 |
| DINOv2 | 52.34 | 51.68 | 40.85 | 40.40 |

## 5.2 EXPERIMENTAL RESULTS

**Main results.** In Table 2, we report the zero-shot image classification accuracy on ImageNet-1k dataset. From the table, we observe that the baselines combined with the proposed RegCache consistently achieves better accuracy in most settings. Specifically, the baselines combined with RegCache outperforms in terms of both best accuracy (Best $\Delta$) and the average accuracy (Average $\Delta$) across the base quantization methods. Only one setup—DINOv2-B—exhibits a negligible accuracy drop.

For zero-shot image–text retrieval (Table 3), we similarly observe thatcombining RegCache with the base quantization methods yields higher performance across all setups on average. The results indicate our method's ability to integrate well with other quantization methods across diverse tasks.

**Reducing token norm outliers.** Table 4 illustrates the change in the maximum token norm of the quantization-sensitive layer input when RegCache is applied. The maximum token norm decreases, while the average of the remaining tokens remains nearly consistent. This reduction effectively narrows the dynamic range of quantization, thereby improving quantization performance.

**Universality of prefixes.** Since the prefix search procedure in RegCache involves validations on the training split of the ImageNet-1K dataset, we also assess whether the learned prefix remains effective on other datasets, as the register token might have overfitted to ImageNet-1K. The results in Table 5 indicate that the prefix from ImageNet-1k remains effective on other datasets, suggesting that it acts as a universal register token.

Table 5: **Zero-shot classification accuracy (%) on other image classification datasets**. The prefixes used in our method are searched using the training split of ImageNet-1K.

| Model | Method | StanfordCars | Flowers-102 | Food-101 | CIFAR-100 |
|-------|--------|-------------|-------------|----------|-----------|
| CLIP-B/16 | FP32 | 64.41 | 65.88 | 85.22 | 68.44 |
| | Naïve | 29.76 | 26.20 | 33.30 | 35.96 |
| | w/ RegCache | 53.14 (+23.38) | 58.53 (+32.33) | 77.38 (+44.08) | 55.27 (+19.31) |
| OpenCLIP-B/16 | FP32 | 88.07 | 69.88 | 83.77 | 76.82 |
| | Naïve | 74.85 | 42.97 | 36.44 | 40.61 |
| | w/ RegCache | 85.96 (+11.11) | 67.86 (+24.89) | 80.97 (+44.53) | 72.26 (+31.65) |
| SigLIP-B/16 | FP32 | 90.81 | 82.63 | 89.34 | 72.33 |
| | Naïve | 87.97 | 75.26 | 78.31 | 54.79 |
| | w/ RegCache | 89.75 (+1.78) | 80.29 (+5.03) | 88.16 (+9.85) | 67.60 (+12.81) |
| SigLIP2-B/16 | FP32 | 92.74 | 83.38 | 90.65 | 77.10 |
| | Naïve | 35.12 | 26.38 | 30.55 | 20.92 |
| | w/ RegCache | 82.59 (+47.47) | 68.29 (+41.91) | 81.83 (+51.28) | 53.92 (+33.00) |

**Ablation study.** In Table 6, we ablate the effect of our two stage methods—(1) prefix caching and (2) token deleting—with quantization performance. The results show that prefix caching suppresses the growth of outliers and improves performance, as token deletion further reduces their magnitude for additional gains. Still, outlier tokens can be safely removed only with prefix caching, as prefix tokens play a similar role to outlier tokens; without it, removing outliers in quantization-sensitive layers severely degrades performance.

**Other experiments.** Besides the main experiments, we further conduct several experiments: (1) Text-Image retrieval results on other vision encoders in Appendix C, (2) Results for weight-only quantization in Appendix D, (3) Computational efficiency of RegCache via FLOPs in Appendix E, and (4) Visualization of searched register tokens in Appendix F.

Table 6: **Ablation studies.** We evaluate each component of RegCache on SigLIP.

| Method | IN-1K acc. |
|--------|-----------|
| Baseline | 69.71 |
| Prefix Caching | 74.21 |
| Token Deleting | 38.51 |
| Prefix Caching + Token Deleting | 74.42 |

## 6 CONCLUSION

In this paper, we have introduced a training-free outlier mitigation algorithm, *RegCache*, designed to enhance the performance of per-tensor post-training quantization (PTQ) for transformer-based vision encoders. Through extensive experiments, we have demonstrated that RegCache consistently improves quantization accuracy when applied on top of existing PTQ methods. Our analysis reveals that RegCache effectively suppresses the activation outliers in quantization-sensitive layers, thereby narrowing the dynamic range and improving quantization stability. Furthermore, this work offers a novel approach to identifying register tokens that are optimal for quantization in vision encoders—a task that is inherently more elusive than in language models.

**Limitations.** A major limitation of our study lies in that the proposed method requires evaluating a number of prefix candidates to identify the most effective configuration. Also, our method introduces several additional hyperparameters to be considered, e.g., the maximum number of token deletions and the number of prefix tokens.

**Discussions and future directions.** Our work covers a variety of vision encoders, including those trained on multimodal data (e.g., CLIP) and those trained on vision-only data (e.g., DINOv2). In our experiments, we observe that quantization-related measures (e.g., quantization sensitivity) behave differently across these cases, warranting further study. Another research direction arises from the differences between LLMs and ViT-based vision encoders: their outlier behavior differs significantly (see Appendix G for an extended discussion). Understanding this phenomenon would benefit wide range of domains, including quantization and representation learning.

REPRODUCIBILITY STATEMENT

To ensure reproducibility, the implementation code is included in the supplementary materials submitted with this paper.

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

# APPENDIX

## A    BASELINES

In this section, we briefly introduce baseline quantization methods for our experiments.

- **PTQ4ViT** (Yuan et al., 2022): Proposes a twin uniform quantizer to handle the unbalanced activation distributions found in ViTs, particularly after non-linearities such as Softmax and GELU.
- **RepQ-ViT** (Li et al., 2023): Addresses quantization bottlenecks by applying specialized preprocessing to sensitive layers, such as channel-wise quantization after LayerNorm and log2 quantization after Softmax.
- **NoisyQuant** (Liu et al., 2023): Introduces a quantizer-agnostic strategy that adds fixed uniform bias to activations, thereby reducing the quantization error of heavy-tailed distributions.

## B    ADDITIONAL RESULTS OF QUANTIZATION SENSITIVITY

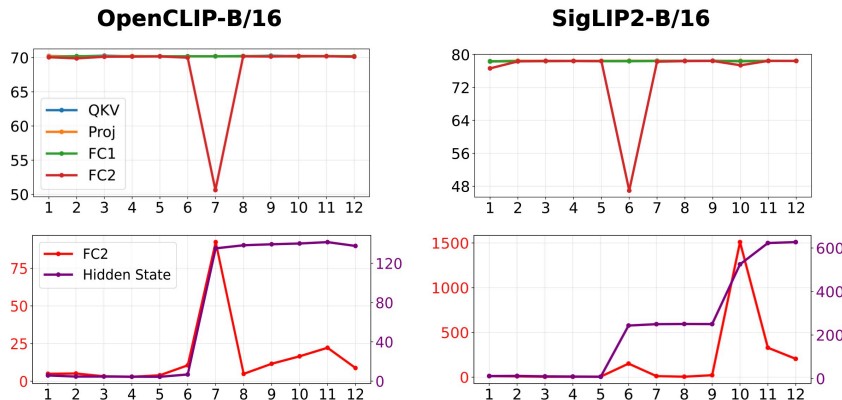

Figure 5: **(Top) Layerwise quantization sensitivity (%).** Zero-shot ImageNet-1K accuracy when we quantize only one layer to W8A8. **(Bottom) Layerwise max token norms.** The largest $\ell_\infty$-norm of all tokens in an image, averaged over the ImageNet-1K validation set.

In this section, we provide additional plots for other vision encoders, analogous to those in Figure 2. In Fig. 5, we plot layerwise quantization sensitivity (top row) and maximum token norm (bottom row) for OpenCLIP and SigLIP2. The trends are consistent with our analysis in Section 3.1: increases in maximum token norm coincide with decreases in quantization sensitivity. However, in the case of SigLIP2, the absolute scale of the maximum norm—both for *fc2* and the hidden state—is significantly larger than in the other architectures we considered. Consequently, applying RegCache yields a clearer benefit, as shown in Table 2. Given SigLIP2's distinct behavior compared to other vision encoders, it would be intriguing to investigate further; we leave this for future work.

## C    IMAGE RETRIEVAL RESULTS

We presents supplementary plots for additional vision encoders, in a format consistent with Table 3. Table 7 presents zero-shot image–text retrieval results on the MSCOCO dataset for OpenCLIP and SigLIP2. In both models, RegCache consistently achieves the highest retrieval accuracy as well.

## D    WEIGHT-ONLY QUANTIZATION

We evaluate the performance of weight-only quantization to assess the compatibility of RegCache with weight-centric methods, which are commonly employed to reduce memory and deployment

Table 7: Zero-shot image–text retrieval performance on MS-COCO, reported as Recall@K (%) for each baseline.

(a) OpenCLIP-B/16

| | I → T | | T → I | |
|---|---|---|---|---|
| | **R@1** | **R@5** | **R@1** | **R@5** |
| FP32 | 61.02 | 83.04 | 41.38 | 66.93 |
| Naïve | 37.32 | 61.12 | 26.30 | 49.38 |
| w/ Ours | 58.00 | 81.08 | 39.00 | 64.48 |
| PTQ4ViT | 59.60 | 82.16 | 40.66 | 66.40 |
| w/ Ours | 59.82 | **82.66** | **41.18** | **66.64** |
| RepQ-ViT | 57.62 | 80.80 | 38.88 | 64.33 |
| w/ Ours | 58.90 | 81.60 | 39.20 | 65.00 |
| NoisyQuant | 57.92 | 80.08 | 39.36 | 64.93 |
| w/ Ours | **60.50** | 82.50 | 40.67 | 66.36 |

(b) SigLIP2-B/16

| | I → T | | T → I | |
|---|---|---|---|---|
| | **R@1** | **R@5** | **R@1** | **R@5** |
| FP32 | 71.60 | 89.16 | 52.33 | 76.58 |
| Naïve | 14.26 | 27.76 | 13.86 | 28.63 |
| w/ Ours | 56.94 | 79.30 | 42.71 | 68.54 |
| PTQ4ViT | 70.10 | 88.66 | 51.35 | 75.56 |
| w/ Ours | **70.82** | **88.74** | **51.64** | **76.27** |
| RepQ-ViT | 69.24 | 87.92 | 50.22 | 74.44 |
| w/ Ours | 69.24 | 87.62 | 50.20 | 74.45 |
| NoisyQuant | 62.28 | 83.50 | 46.39 | 71.43 |
| w/ Ours | 62.86 | 84.50 | 46.30 | 71.45 |

Table 8: Zero-shot image classification accuracy (%) under weight-only quantization (AWQ), comparing our method with the baseline.

| Model | FP32 | Method | Weight-only (AWQ) Bits | | | |
|---|---|---|---|---|---|---|
| | | | **W8A32** | **W6A32** | **W4A32** | **W3A32** |
| **CLIP-B/16** | 68.32 | AWQ | 68.23 | 68.18 | 67.83 | 65.90 |
| | | + RegCache | 68.27 (+0.04) | 68.25 (+0.07) | 67.95 (+0.12) | 66.14 (+0.24) |
| **OpenCLIP-B/16** | 70.22 | AWQ | 69.39 | 69.38 | 68.99 | 66.72 |
| | | + RegCache | 69.40 (+0.01) | 69.40 (+0.02) | 68.97 (-0.02) | 66.80 (+0.08) |
| **SigLIP-B/16** | 76.05 | AWQ | 75.72 | 75.89 | 75.39 | 72.34 |
| | | + RegCache | 75.79 (+0.07) | 75.94 (+0.05) | 75.43 (+0.04) | 72.48 (+0.14) |
| **SigLIP2-B/16** | 78.48 | AWQ | 77.13 | 77.14 | 76.46 | 71.51 |
| | | + RegCache | 77.32 (+0.19) | 77.33 (+0.19) | 76.65 (+0.19) | 71.75 (+0.24) |

cost. Specifically, we adopt AWQ (Lin et al., 2024)—a popularly adopted weight-only quantization algorithm—as the base method, using a group size of 128 and varying bitwidths of 8, 6, 4, and 3. Across all configurations, RegCache consistently improves performance over vanilla AWQ, demonstrating its complementary benefit even in memory-constrained quantization settings.

# E COMPUTATIONAL EFFICIENCY OF REGCACHE

Table 9: **Comparison of GFLOPs.** The best configuration selected by our method in RegCache.

| Model | Vanilla | RegCache | Δ (%) |
|---|---|---|---|
| CLIP | 35.26 | 35.33 | +0.20 |
| OpenCLIP | 35.25 | 35.31 | +0.17 |
| SigLIP | 35.53 | 35.26 | -0.76 |
| SigLIP2 | 35.53 | 35.47 | -0.17 |
| DINOv2 | 46.46 | 46.47 | +0.02 |

In terms of computational efficiency, we quantitatively assess the computational overhead induced by the proposed RegCache method by measuring the FLOPs[3]. As shown in Table 9, the increase in FLOPs due to RegCache remains minimal—no more than 0.2%p of the total computation.

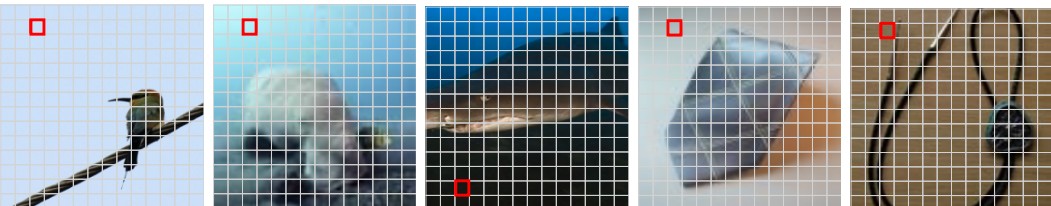

Figure 6: Visualization of register token searched from ImageNet-1K

## F VISUALIZATION OF REGISTER TOKENS

In Fig. 6, we visualize the top 5 prefix tokens ranked by their effectiveness, as measured by zero-shot classification accuracy on ImageNet-1k. The results are consistent with prior findings (Darcet et al., 2024; Jiang et al., 2025), revealing that the searched register tokens are located in background regions. We find that the searched register tokens commonly correspond to low-frequency regions surrounded by semantically uninformative patches.

## G OUTLIERS IN VISION ENCODER VS. LLMS: A TOKENIZATION PERSPECTIVE

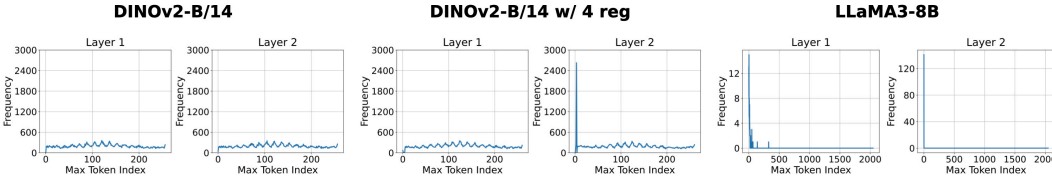

Figure 7: **The frequency of top-1 max tokens in the input tensor of FC2 layers in different models**. We evaluated DINOv2 on ImageNet-1K and LLaMA3-8B on Wikitext-2 dataset.

As shown in Fig. 2, various vision encoders exhibit outliers in the intermediate layers. Rewinding recent studies about outlier tendency in LLMs (Sun et al., 2024; Gu et al., 2025), it is natural to ask: why do outliers consistently emerge in the intermediate layers of vision encoders, rather than in the early layers as in LLMs? In this section, we investigate this phenomenon by rethinking the difference of *tokenization strategies* between vision encoders and LLMs.

Roughly speaking, LLMs map input sequences to tokens drawn from a discrete, fixed vocabulary. In contrast, vision encoders and other ViT-based models process inputs by mapping them to continuous token embeddings using a (convolutional) neural network. We hypothesize that differences in the emergence of sink tokens can be attributed from their fundamentally distinct tokenization process.

To test this hypothesis, we compare the outlier behavior of DINOv2, pretrained both with and without learned register tokens, and LLaMA3-8B (Dubey et al., 2024). In this setup, the register tokens act as "fixed outlier sinks," (Darcet et al., 2024) effectively forming a closed-set vocabulary for outlier attraction, analogous to the tokenization setup in LLMs. As shown in Fig. 7, when ViTs are equipped with four learned register tokens, they begin to exhibit outliers in early layers (i.e., 2nd layer), mirroring the behavior observed in LLMs. This supports our hypothesis that continuous tokenization in ViTs plays a crucial role in the emergence of outliers in the intermediate layers.

## H LLM USAGE

We used a large language model to help refine the writing of this paper.

---

[3]FLOPs are estimated using a pseudo-quantization approach that simulates quantized operations.

