# OpenReview forum: "Post-training quantization of vision encoders needs prefixing registers"
_ICLR.cc/2026/Conference — ICLR 2026 Conference Withdrawn Submission_

### Official Review · Reviewer_TDJs · 2025-10-16

**Soundness:** 2
**Presentation:** 2
**Contribution:** 2
**Rating:** 2
**Confidence:** 3

**Summary:**

This paper proposes RegCache, a method to improve post-training quantization of vision encoders by using prefixing registers to handle activation outliers. It claims to be training-free and shows improved results across various models and quantization techniques.

**Strengths:**

- Interesting observation about middle-layer outlier emergence in vision encoders

- Clear empirical demonstration of performance improvements

- Comprehensive evaluation across multiple vision encoders (CLIP, DINOv2, SigLIP, etc.) and quantization methods

**Weaknesses:**

- The method tunes the number of tokens to delete (k̃) based on downstream performance.
This process constitutes a form of task-specific, validation-based tuning that is computationally expensive and requires labeled data from the target domain. This violates the standard premise of PTQ, which is designed to be a lightweight process that does not require access to the final task's labels or metric.

- The approach inserts external tokens as pre-computed KV at middle layers and removes tokens deemed sinks. This goes beyond calibration as it alters the attention context and risks semantic/architectural drift.

**Questions:**

- What is the computational cost of the entire RegCache pipeline, including the candidate curation and hyperparameter search? How does this cost compare to simply performing a brief Quantization-Aware Training (QAT) round, which would likely be more effective?

- The paper compares against other PTQ methods. Were these baselines also allowed to perform a similar level of task-specific hyperparameter tuning on the ImageNet validation set? If not, the comparison is not fair.

---

> ### Author Response · Authors · 2025-11-17
>
> ### **W1&Q1. RegCache as a “non-standard” PTQ method & Computational cost vs. QAT**
>
> >The method tunes the number of tokens to delete (k̃) based on downstream performance. This process constitutes a form of task-specific, validation-based tuning that is computationally expensive and requires labeled data from the target domain. This violates the standard premise of PTQ, which is designed to be a lightweight process that does not require access to the final task's labels or metric.
>
> >What is the computational cost of the entire RegCache pipeline, including the candidate curation and hyperparameter search? How does this cost compare to simply performing a brief Quantization-Aware Training (QAT) round, which would likely be more effective?
>
> We appreciate the reviewer’s careful feedback and acknowledge that our method does not fully match the strictest definition of PTQ. However, RegCache requires only inference-time GPU memory, which is substantially smaller than that needed for QAT. Although it uses supervision data and introduces a slightly higher computational cost than existing calibration-based PTQ methods, this overhead remains modest and practically manageable. For more description, RegCache is not intended to replace existing PTQ algorithms, but as a **plug-and-play module** that can be applied on top of them.
>
> Additionally, **our procedure does not require final-task labels and also do not tune *k* for every dataset or task.** Instead, we determine *k* once per model (or per baseline) using ImageNet-1k zero-shot classification as a single, general-purpose reference task, and reuse the same value across all downstream datasets and tasks. Despite this minimal tuning, we consistently observe clear improvements over all baseline PTQ methods, as shown in Tables 3 and 5.
>
> We will make this point more explicit in the revised manuscript.
>
> ---
>
> ### **W2. Risk for altering attention context**
> >The approach inserts external tokens as pre-computed KV at middle layers and removes tokens deemed sinks. This goes beyond calibration as it alters the attention context and risks semantic/architectural drift.
>
> The concern regarding potential semantic or architectural drift appears less applicable in this setting. Prior work suggests that the specific tokens we target behave more like attention bias/noise rather than semantically meaningful content.
>
> Several recent studies on transformers have shown that (1) attention sinks / register-like tokens often carry little semantic information, acting as structural biases that attract attention regardless of input content [1,2,3], and (2) even such sinks have been linked to degraded interpretability [4].
>
> In particular, prior work on register tokens in vision transformers demonstrates that introducing such tokens can stabilize representations without degrading semantic performance and, even improve downstream accuracy by mitigating sink tokens that effectively act as noise in the attention maps. [2, 4]
>
> ---
>
> ### **Q2. Fairness of comparisons vs. other PTQ methods**
> >The paper compares against other PTQ methods. Were these baselines also allowed to perform a similar level of task-specific hyperparameter tuning on the ImageNet validation set? If not, the comparison is not fair.
>
> First, we clarify that we do not directly compare RegCache against other PTQ methods. Our goal is to present RegCache as a broadly complementary algorithm to existing PTQ methods (as described in Sec. 5.1), not as a standalone competitor. Because the comparison is not a direct “method A vs. method B,” the concern about equal tuning budgets does not directly apply.
>
> &nbsp;
>
> [1] Sun, Mingjie, et al., Massive activations in large language models, COLM 2024
>
> [2] Jiang, Nick, et al., Vision Transformers Don't Need Trained Registers, NeurIPS 2025
>
> [3] Kang, Seil, et al., See what you are told: Visual attention sink in large multimodal models, ICLR 2025
>
> [4] Lu, Andrew, et al., Artifacts and attention sinks: Structured approximations for efficient vision transformers, arXiv 2025

---

### Official Review · Reviewer_u2Yn · 2025-10-29

**Soundness:** 2
**Presentation:** 3
**Contribution:** 2
**Rating:** 2
**Confidence:** 3

**Summary:**

RegCache is a training-free method to reduce inference cost in transformer-based vision encoders (e.g., CLIP) by mitigating activation outliers that hinder 8-bit post-training quantization. It injects semantically meaningless, outlier-prone prefix tokens to prevent other tokens from producing outliers and, based on the distinct behavior of vision encoders vs. language models, introduces two key techniques: middle-layer prefixing and token deletion. This enables quantization with significantly smaller accuracy drops and consistently improves performance across both text-supervised and self-supervised encoders.

**Strengths:**

1. This paper is well organized and easy to follow.
2. The method is plug-and-play.
3. The authors provide code for reproducing.

**Weaknesses:**

2. There are lots of works that propose to address the issue of outliers for LLM (which can also apply to ViT) and ViT. However, this paper does not compare the proposed method with them. For example, QuaRot (NeurIPS 2024), https://openreview.net/pdf?id=Uh5XN9d2J4 (ICML 2024), etc.
3. This paper does not implement their method on top of the SOTA quantization methods to show the effectiveness of their approaches. For example, FIMA-Q (CVPR 2025) and APHQ-ViT (CVPR 2025).
4. The introduced overhead of this method for real-time inference is not being discussed.
5. The idea that introducing additional tokens to mitigate outliers and sinks is not novel (see https://arxiv.org/abs/2402.17762 and https://arxiv.org/abs/2410.05265).

**Questions:**

N/A

---

> ### Author Response · Authors · 2025-11-17
>
> ### **Q1. Experiment for QuaRot**
> >There are lots of works that propose to address the issue of outliers for LLM (which can also apply to ViT) and ViT. However, this paper does not compare the proposed method with them. For example, QuaRot (NeurIPS 2024), https://openreview.net/pdf?id=Uh5XN9d2J4 (ICML 2024), etc.
>
> RegCache is a plug-in module that can be implemented on top of QuaRot. Our approach is complementary to rotation-based outlier mitigation methods, rather than overlapping with them. We will include the updated results in the revised version. Additionally, since Ma et al. did not release their code, we are currently in contact with them to request it.
>
> ---
>
> ### **Q2. Experiment on FIMA-Q**
> >This paper does not implement their method on top of the SOTA quantization methods to show the effectiveness of their approaches. For example, FIMA-Q (CVPR 2025) and APHQ-ViT (CVPR 2025).
>
> As suggested, we implemented experiments applying RegCache on top of FIMA-Q. Across all models and bit-widths, RegCache consistently improves performance over FIMA-Q. Notably, in the 4-bit setting, our method improves CLIP’s accuracy from 50.41% to 62.08%, establishing a new state of the art. This demonstrates that activation-outlier suppression is also complementary to the round-function optimization used in PTQ methods.
>
> ---
>
> ### **Q3. Wall-clock overhead**
> >The introduced overhead of this method for real-time inference is not being discussed.
>
> In response to your comment, we have conducted experiments to measure the inference-time overhead introduced by RegCache. Specifically, we report wall-clock latency for ImageNet-1k validation within vanilla W8A8 PTQ and for W8A8 combined with RegCache under the pseudo-quantization setup. The results are summarized in the table below.
>
> #### **Wall-clock time**
> | Model        | Vanilla | RegCache | Delta (%) |
> |:--------------|:------------:|:----------:|:-----------:|
> | SigLIP-B/16  | 1m 43.574s  | 1m 43.749s   | 0.17       |
>
> As a result, we observe that RegCache introduces only a small additional latency compared to vanilla W8A8.
>
> ---
>
> ### **Q4. Lack of novelty: mitigate outliers with sink-like tokens**
> >The idea that introducing additional tokens to mitigate outliers and sinks is not novel (see https://arxiv.org/abs/2402.17762 and https://arxiv.org/abs/2410.05265).
>
> We respectfully disagree with the reviewer's comment that our work lacks novelty. Our work differs from existing sink-based approaches in both problem setting and mechanism.
>
> While previous work focuses entirely on LLMs and relies on language-specific properties that do not transfer to vision encoders, the situation is fundamentally different for vision encoders (which are ViT-variants). As shown in Figure 1, unlike LLMs—where register tokens can be directly identified in the discrete input embedding layer—**It is difficult to identify register tokens at the input stage of ViTs because they operate on continuous embedding vectors.**
>
> RegCache goes beyond these observations by (1) establishing a generalizable property—high cross-image cosine similarity—that enables the identification of register tokens within continuous embeddings, and (2) developing a method for extracting prefix tokens that effectively mitigate outliers, rather than stopping at the observational level as prior work does.

---

### Official Review · Reviewer_JjR7 · 2025-10-31

**Soundness:** 2
**Presentation:** 2
**Contribution:** 3
**Rating:** 4
**Confidence:** 4

**Summary:**

This paper proposes the RegCache algorithm to address the outlier problem in post-training quantization (PTQ) of vision encoders. Specifically, RegCache introduces outlier-prone yet semantically meaningless prefix tokens to the target vision encoder, which prevents other tokens from having outliers. This design ultimately reduces the quantization error of PTQ.

**Strengths:**

The analysis of outliers in PTQ quantization is interesting, and its approach to reducing model outliers is innovative, which merits further research.

**Weaknesses:**

The paper proposes several key hypotheses and observations that underpin its RegCache method, yet these claims undermine the work’s generality and reliability. They are not sufficiently supported by either extensive experimental validation (e.g., across more diverse image domains or model architectures) or in-depth theoretical analysis—leaving their applicability to vision encoder post-training quantization (PTQ) insufficiently verified.

**Questions:**

1. The paper lacks theoretical analysis for key observations—specifically, it fails to provide a theoretical explanation for the intrinsic reason behind the "cross-image similarity of middle-layer outliers". For instance, an analysis from perspectives such as the attention mechanism or LayerNorm is absent.
2. The paper has issues with its figures. The left subfigure of Figure 1 does not provide sufficiently detailed information. The right subfigure of Figure 1 lacks adequate descriptions of experimental setup details. Figure 2 is deficient in introductions to key terms.
3. Although the authors cite numerous references, the paper still needs to explain in the text why sink tokens can mitigate outliers.
4. Provide a detailed comparison between RegCache and recent outlier quantization methods for vision encoders.
5. Supplement quantitative experiments to analyze the proportion of quantization error contributed by outliers, so as to further support the urgency of the research motivation.

---

> ### Author Response · Authors · 2025-11-17
>
> ### **W1. Generalizability across datasets and models**
> > The paper proposes several key hypotheses and observations that underpin its RegCache method, yet these claims undermine the work’s generality and reliability. They are not sufficiently supported by either extensive experimental validation (e.g., across more diverse image domains or model architectures) or in-depth theoretical analysis—leaving their applicability to vision encoder post-training quantization (PTQ) insufficiently verified.
>
> First, we clarify that we already experiment on extensive setups, including:
>
> - Wide range of foundation vision encoders (CLIP, OpenCLIP, SigLIP, SIGLIP 2,  and DINOv2), for which previous work did not address for quantization
> - Wide range of datasets (ImageNet, MS-COCO retrieval, Stanford Cars, Flowers-102, Food-101, and CIFAR-100)
>
> We would appreciate any further suggestions if the reviewer has any specific setting in mind.
> Regarding theoretical analysis, this is mainly due to a shortage of theoretical explanations on why attention sink tokens emerge. Thus we have taken a rather empirical approach. Nevertheless, we will add more discussions on this point.
>
> ---
>
> ### **Q1. Theoretical perspective on high cosine similarity of outlier tokens**
> >The paper lacks theoretical analysis for key observations—specifically, it fails to provide a theoretical explanation for the intrinsic reason behind the "cross-image similarity of middle-layer outliers". For instance, an analysis from perspectives such as the attention mechanism or LayerNorm is absent.
>
> Our empirical observation is that outlier tokens in mid-layers tend to occur in a small number of channels in feature space, which leads to high cosine similarity across images. This phenomenon is consistent with prior work [1,2], showing that activation outliers in large transformers frequently concentrate on a small subset of channels, and these channel-wise outliers recur across inputs.
> In the revised version, we explicitly update paragraphs explaining this intuition.

---

> ### Author Response · Authors · 2025-11-17
>
> ### **Q3. Why register tokens mitigate attention sink and activation outliers**
> > Although the authors cite numerous references, the paper still needs to explain in the text why sink tokens can mitigate outliers.
>
> During self-attention, register tokens receive consistently high attention weights since they exhibit sink-like behavior. Therefore, their presence can **redistribute the attention weights under the softmax mechanism** and reduce the dominant weights of sink tokens [1].
>
> Moreover, as several prior studies have reported [1,3], attention sinks are coupled with the emergence of activation outliers in transformer models. Building on this insight, we mitigate these attention sinks by introducing cached register tokens, thereby reducing the norm of activation outliers that cause dominant quantization errors.
>
> ---
>
> ### **Q4. Comparison with recent ViT PTQ methods**
> > Provide a detailed comparison between RegCache and recent outlier quantization methods for vision encoders.
>
> In a nutshell, RegCache is the first plug-and-play module that directly mitigates activation outliers through internal register tokens. Our method is designed to be seamlessly combined with existing PTQ methods for vision encoders, resulting in better performance when RegCache is applied on top of them.
>
> Recent methods that address activation outliers in transformer-based models include **FIMA-Q** [4], which optimizes the rounding function for PTQ, and **QuaRot** [5], which applies a rotation matrix to reduce outlier magnitude.
> We are conducting experiments based on these algorithms, and the results show positive effect on quantization performance.
>
> We will include these experiments using these baselines in the next revised version.
>
> ---
>
> ### **Q5. Quantization error attributed to outliers**
> >Supplement quantitative experiments to analyze the proportion of quantization error contributed by outliers, so as to further support the urgency of the research motivation.
>
> As suggested, we conducted experiments comparing the quantization of the entire tensor with an approach that quantizes outliers separately using a token-wise quantizer. We identify outlier tokens in each activation tensor based on a norm threshold (i.e., tokens whose norms are more than 3-$\sigma$ above the mean token norm), which is a standard criterion in the quantization literature. The result below shows the accuracy of zero-shot classification on ImageNet-1k.
>
> #### **W8A8 Results (Acc.)**
> | Model         | Full Precision | Full Tensor | Outlier Separated |
> |:---------------|:----------------:|:-------------:|:--------------------:|
> | CLIP-B/16     | 68.32           | 34.01       | 55.83              |
> | OpenCLIP-B/16 | 70.22          | 46.12       | 65.43              |
> | SigLIP-B/16   | 76.05          | 69.71       | 74.54              |
> | SigLIP2-B/16  | 78.47          | 26.04       | 74.54              |
> | DIOV2-B/16    | 83.26          | 19.20       | 76.58              |
>
> We found that outlier tokens contribute a disproportionately large fraction of the total quantization error.
>
> ---
>
> ### **Other comments.**
> We thank the reviewer for this careful feedback about figures. We agree that our initial figures lacked sufficient detail for full clarity and will revise the manuscript accordingly.
>
> &nbsp;
>
> [1] Sun, Mingjie, et al., Massive activations in large language models, COLM 2024
>
> [2] Jiang, Nick, et al., Vision Transformers Don't Need Trained Registers, NeurIPS 2025
>
> [3] Gu, Xiangming, et al. When attention sink emerges in language models: An empirical view, ICLR 2025
>
> [4] Wu, Zhuguanyu, et al., FIMA-Q: Post-Training Quantization for Vision Transformers by Fisher Information Matrix Approximation, CVPR 2025
>
> [5] Ashkboos, Saleh, et al., QuaRot: Outlier-free 4-bit inference in rotated llms, NeurIPS 2024

---

### Official Review · Reviewer_d41b · 2025-11-05

**Soundness:** 2
**Presentation:** 2
**Contribution:** 2
**Rating:** 4
**Confidence:** 5

**Summary:**

This paper proposed a novel training-free algorithm to mitigate outliers in post-training quantization of vision encoders through pruning semantically meaningless prefix tokens. Experiments reveal the effectiveness of the proposed methods.

**Strengths:**

1. The discovery of semantically meaningless prefix tokens is enlightening.
2. The motivations and corresponding methods are clearly revealed and discussed.

**Weaknesses:**

Major:
1. In Line 96-98, the prefixed tokens are only applied for middle-to-final layers. Is there any experimental comparison and corresponding discussion about how the layer settings effect the quantization process and model performance?
2. In section 3.1, why only layer-wise sensitivities are discussed, since the sensitivities and outliers are in transformers are about channel level or even token level as discussed in previous arts.
3. How to deal with vision encoders like NaViT, which process images as any arbitrary resolution. Thus in these vision encoder pipeline, the prefix can be varied and the quantization strategy proposed in this paper may not perform well.


Minor:
1. It would be better to add legends in Figure 2 for more clarity.
2. Maybe section 3.1 and 3.3 can be combined as one section about how layer index affect the outlier and quantization performance, for better writing logic.

**Questions:**

See weaknesses. My major concern is the third point in the "Major" part.

---

> ### Author Response · Authors · 2025-11-17
>
> ### **W1. The reason for choosing middle-to-final layers and the effect of layer settings**
> > In Line 96-98, the prefixed tokens are only applied for middle-to-final layers. Is there any experimental comparison and corresponding discussion about how the layer settings effect the quantization process and model performance?
>
> Following the reviewer’s suggestion, we have conducted an additional ablation study on which layer to begin caching. Results suggest that the **proposed middle-to-final layer caching consistently outperforms full-layer caching** on all setups considered. We believe this is due to the detrimental effects of adding unnecessary tokens (which are not clearly register-ish) to the early layers.
> We will add the results in the revised manuscript.
>
> ---
>
> ### **W2. The reason for focusing on layer-wise sensitivity**
> >In section 3.1, why only layer-wise sensitivities are discussed, since the sensitivities and outliers are in transformers are about channel level or even token level as discussed in previous arts.
>
> This is because our goal is to develop a prefix-based outlier mitigation method, for which the layerwise sensitivity turned out to be insightful and effective. We refrained from providing analysis for the sake of analysis—especially, we deliberately avoided motivating channel-level algorithms which are difficult to be used for accelerating the model without additional hardware-level considerations. Similarly, token-level analyses are not described in text, as we focus on the per-tensor quantization algorithms.
>
> ---
>
> ### **W3. Applying our method to NaViT and arbitrary-resolution encoders**
> >How to deal with vision encoders like NaViT, which process images as any arbitrary resolution. Thus in these vision encoder pipeline, the prefix can be varied and the quantization strategy proposed in this paper may not perform well.
>
> The primary goal of this paper is to propose an effective activation quantization algorithm for large-scale pre-trained vision encoders, which are widely adopted in modern ML pipelines such as VLM or VLA. Thus we have focused on CLIP, SigLIP, DINO families—unfortunately, there seems to be no open-weight model checkpoints of NaViT-based foundation models (at least with similar popularity).
>
> ---
>
> ### **Other comments.**
>
> We thank the reviewers for helpful suggestions regarding figure and writing. We will incorporate these.

---

### Author Response · Authors · 2025-11-17
**Common Response**

Dear reviewers and the AC,

Thank you for taking your time to review our manuscript.

Although we have decided to ***withdraw*** our manuscript, we would like to take this chance to briefly address some of your concerns. This is not only to clarify some misunderstandings, but also our sincere acknowledgment of your efforts to improve our manuscript—which we deeply appreciate.

We have responded to your comments one-by-one.

Best regards,
Authors.

---

### Note · Authors · 2025-11-17

I have read and agree with the venue's withdrawal policy on behalf of myself and my co-authors.